# Defining Coastal Resilience

**Gerd Masselink [1],\* and Eli D Lazarus [2],\*** 

[1]  Coastal Processes Research Group, School of Biological and Marine Sciences, University of Plymouth, Plymouth PL4 8AA, UK

[2]  Environmental Dynamics Lab, School of Geography & Environmental Science, University of Southampton, Southampton SO17 1BJ, UK

\*  Correspondence: g.masselink@plymouth.ac.uk (G.M.); e.d.lazarus@soton.ac.uk (E.D.L.)

**Abstract:** The concept of resilience has taken root in the discourse of environmental management, especially regarding Building with Nature strategies for embedding natural physical and ecological dynamics into engineered interventions in developed coastal zones. Resilience is seen as a desirable quality, and coastal management policy and practice are increasingly aimed at maximising it. Despite its ubiquity, resilience remains ambiguous and poorly defined in management contexts. What is coastal resilience? And what does it mean in settings where natural environmental dynamics have been supplanted by human-dominated systems? Here, we revisit the complexities of coastal resilience as a concept, a term, and a prospective goal for environmental management. We consider examples of resilience in natural and built coastal environments, and offer a revised, formal definition of coastal resilience with a holistic scope and emphasis on systemic functionality: "Coastal resilience is the capacity of the socioeconomic and natural systems in the coastal environment to cope with disturbances, induced by factors such as sea level rise, extreme events and human impacts, by adapting whilst maintaining their essential functions." Against a backdrop of climate change impacts, achieving both socioeconomic and natural resilience in coastal environments in the long-term (>50 years) is very costly. Cost trade-offs among management aims and objectives mean that enhancement of socioeconomic resilience typically comes at the expense of natural resilience, and vice versa. We suggest that for practical purposes, optimising resilience might be a more realistic goal of coastal zone management.

**Keywords:** coastal management; adaptation; coastal impact of climate change; coastal engineering; nature-based solutions

## 1. Introduction

Coastal environments are among the most intensively used regions of the Earth for supporting human population, activity, and industry [1]. Because this intensive use tends to come at the expense of natural coastal environmental systems, driving ecological and landscape degradation or destruction, the challenge for coastal management is to sustainably balance the fundamental functional needs of human and natural coastal systems for the present and future. In management contexts, coastal resilience is now a keystone concept [2,3] and fundamental to Building with Nature strategies [4] to reduce coastal risk and environmental degradation. The prominence of the resilience concept is pressed to the fore by rapid rates of growth in coastal megacities around the world [5]; by record-setting damage from disaster events such as Hurricanes Katrina (2005), Sandy (2012), and Harvey (2017) in the United States [6] and the winter storms of 2013–2014 and 2015–2016 in the United Kingdom [7,8]; and by the untenable costs of supporting conventional grey infrastructure to protect against coastal hazards [9–14].

However, ambiguity pervades the rapidly growing academic literature that invokes resilience. Scholars who have tracked the term in environmental literature suggest that resilience is trending toward becoming a buzzword devoid of meaning, both amorphous and overused [15–17]. Contributions to the literature are not always specific about what they intend resilience to convey, whether a conceptual reference to patterns of change within a system, a specific property of a system that can be observed or estimated, or a goal to achieve through managed decision-making [18,19]. Some argue that coastal resilience means little without a clearly defined spatial and temporal framework [20].

The ambiguity that freights coastal resilience is a consequence of the many definitions, applications, and adaptations that have proliferated across and within disciplines since the origin of resilience as a theory in ecology [15,21,22]. Resilience thinking [23,24] is now firmly embedded in natural hazards research [18,25], in the study of environmental and social impacts of climate change [26,27], and in discourses of economic and political systems more broadly [28,29]. Resilience now connotes a variety of physical, social, and socioeconomic dimensions, as well as links to explicitly or implicitly related concepts such as vulnerability, sensitivity, susceptibility, persistence, equilibrium, stability, thresholds, tipping points, regime shifts, recovery, adaptive capacity and sustainability [17,30]—many of which contend with their own multiple working definitions and diffuse associations [31]. When adjectives like ecological and engineering—or others, like morphological and socioeconomic—appear beside resilience, they typically refer to the system under consideration, not the type of resilience [32] being invoked.

Here, in an effort to disentangle the various strands of coastal resilience, we revisit the complexities of coastal resilience as a concept, a term, and a prospective goal for environmental management. We consider examples of resilience in natural and built coastal environments, and offer a revised, formal definition of coastal resilience with a holistic scope and emphasis on systemic functionality.

## 2. Origins of Resilience Theory

Resilience theory arose from the study of population fluctuations in ecological systems. Holling [21] proposed that the dynamical behaviour of ecological systems could well be defined by two distinct properties: resilience and stability. Resilience originally referred to the persistence of relationships within a system, a measure of the system's ability to absorb environmental changes with its internal dynamics intact. Stability represented the ability of a system to return to an equilibrium state after a temporary disturbance; the more rapid the return, the more stable the system is. (Consider the stability of a tightly coiled spring—stretch it out and release it, and the spring will snap back to its resting coiled state.) Testament to the convolutions of resilience theory in the decades since its appearance, the original definition of stability is typical of the way resilience is now formalised; that is, the ability to recover or bounce back from a disturbance is now all but synonymous with resilience.

Holling [32] further divided resilience into two types: ecological and engineering resilience, which map onto the original definitions of resilience and stability, respectively [21]. Ecological resilience focuses on persistence, change, and unpredictability, emphasising conditions that drive system dynamics away from any equilibrium steady state, including dynamical instabilities that can flip a system into another regime of behaviour. In the language of dynamical systems, a condition to which a system tends to evolve, for a wide variety of initial conditions, is called an attractor [33]. Ecological resilience acknowledges the existence of multiple potential equilibria—multiple dynamical attractors—and so is defined as the amount of disturbance that a system can sustain before undergoing a fundamental change in controls and structural organisation. By comparison, engineering resilience focuses on efficiency, consistency, and predictability, emphasising conditions that facilitate system stability around a single, global equilibrium steady state (a single, dominant dynamical attractor). Resistance to disturbance and the rate of return to the equilibrium condition—both derived from classical considerations of stability in engineering and economics—are used as measures of engineering resilience. Ecological and engineering resilience are less mutually exclusive than they are end-members of a resilience continuum. An ecological system might exhibit degrees of resistance to disturbance—a

property of engineering resilience—while also possessing the capacity to reorganize into another state if disturbance exceeds a critical threshold—a property of ecological resilience [34].

## 3. Resilience in Natural Coastal Environments

Understanding controls on landscape resilience, and how ecosystems and landscapes coevolve, are two closely related grand challenges in geomorphology [35]. Shaped by feedbacks between fluid flow, sediment transport, ecology, and changeable morphology, coastal environments showcase a remarkable variety of settings in which to explore both of these open questions. Steady-state and dynamic equilibrium behaviours in geomorphic systems require resilience to dampen out fluctuations and retain what manifests as long-term stability. Geomorphologists tend to invoke the engineering definition of resilience, emphasising consistency and predictability, perhaps because the concept of long-term steady-state conditions is so close to the core of the traditional discipline [16]. However, when a geomorphic system does not recover from a perturbation—when a driver is cut off or an internal threshold has been exceeded—and enters a different, perhaps equally persistent state, this transition represents a form of ecological resilience, characterised by the presence of multi-stable states. Indeed, where geomorphology is considered a physical determinant of ecosystem resilience, the definition of ecological resilience is most widely used [14]. Alternative stable states, and dynamical transitions between them, have been more extensively explored for ecology and ecosystems [36–38] than for geomorphology [39], but multiple or alternative stable states are a common characteristic of coastal landscapes [40,41].

### 3.1. Barrier Islands And Beaches

Barrier islands are considered exemplars of coastal resilience [42] (Figure 1). Coastal barriers are landforms that tend to maintain their height and cross-shore width even as they transgress landward over time [43–46]. Their response to short-term storm impacts, in which overwash flow transfers sediment from the foreshore to the back barrier, is what ultimately sustains their morphology over extended timescales [47]. According to Long et al. [20], large barrier systems are inherently resilient landforms as long as they are able to internally recycle sediment to maintain overall landform integrity. Stéphan et al. [48] contend that, as long as the rate of sea level rise is not excessive and there is no sediment deficit, barrier systems are surprisingly resilient, even to the most extreme storm events. Beach dynamics appear to describe an oscillating attractor in response to seasonal storm events, with at least two morphological regimes (narrow and wide, or reflective and dissipative) over multiannual to decadal timescales [49–53], likely driven by large-scale ocean–atmospheric patterns [54]. Beaches erode during storms and recover under calmer wave conditions and the ability of a beach to recover from storm erosion is clearly an expression of resilience [55]. The more rapid recovery of beaches compared to that of coastal dunes, suggests perhaps that beaches are more resilient to storm impacts than dunes [56]. Resiliency of a barrier beach may be dependent on the rate of post-storm dune recovery; for locations with a relatively long recovery period (>10 years), a change in storm magnitude and/or frequency is a potential threat to barrier island resilience [57].

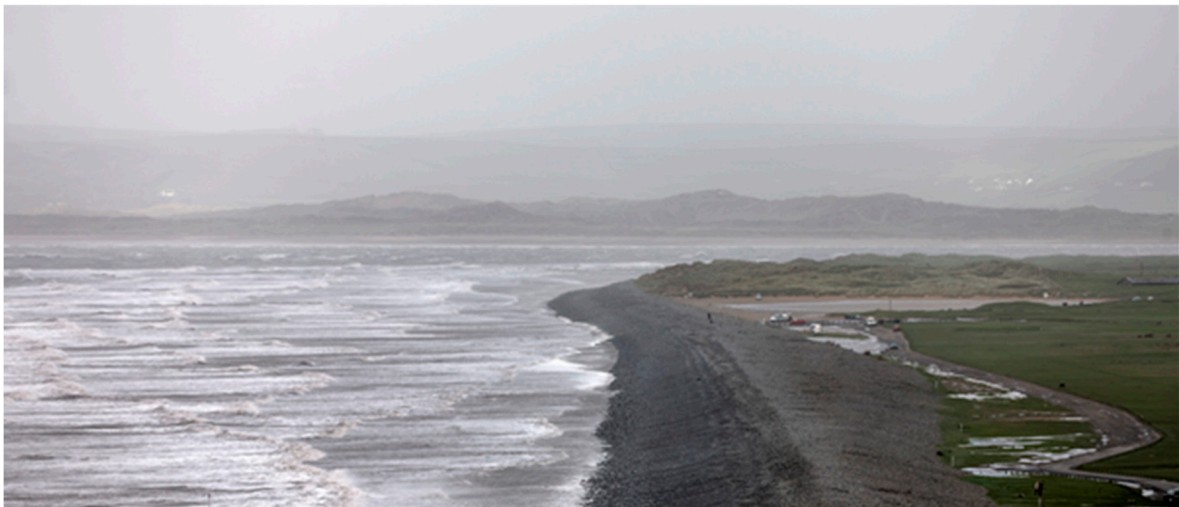

**Figure 1.** Gravel barriers are natural forms of coastal defence that protect the hinterland from flooding, whilst at the same time being able to respond to sea level rise and extreme storms by rolling back through overtopping and washover processes (photo: Gerd Masselink). They are thus exemplars of a coastal landform resilient to both pulse and ramp disturbances.

### 3.2. Coastal Dunes

Coastal dunes grow as a result of coupled interactions between marine and aeolian forcing [58,59], and through a feedback between vegetation and sediment transport, in which shallow burial promotes plant growth that enhances further sediment deposition [60–63]. Barrier dunes express two end-member states—low and high—that are sensitive to vegetation as a control on sediment-transport pathways and storage [64–67]. As storm impacts erode dunes and aeolian processes construct them, both alternative states of high and low dunes can exist in space immediately adjacent to each other, with dune vegetation serving to both resist storm-driven flattening and augment dune growth by trapping windblown sediment [68,69]. A low, overwash-reinforcing state [64] exhibits a weakly positive sediment budget, burial-tolerant grasses, flat topography, and frequent overwash. A high, overwash-resisting state exhibits a strongly positive sediment budget, burial-intolerant grasses, ridge and swale topography, and infrequent overwash. In each domain, plant adaptations exert an influence on external variability by shaping topographic recovery in a way that reinforces the conditions and overwash exposures for which they are better adapted [60–62]. These feedbacks and their domain states can vary within an individual island and among adjacent islands [70].

### 3.3. Tidal Wetlands

Much like in dune systems, a similar feedback between vegetation and sedimentation sustains tidal wetlands, such as salt marshes and mangroves, enabling them to maintain their elevations relative to sea level [71–73]: a slightly deeper tidal prism (forced by sea level rise) carries more fine sediment in suspension; tidal–wetland vegetation slows flow velocity, causing sediment deposition that the presence of vegetation helps trap in place; and shallow burial and nutrient delivery promotes biomass growth above and below ground, driving a net increase in platform elevation. Sediment supply is a key factor in salt marsh resilience [74,75]. Storms play a key role in the response of salt marshes to sea level rise, but salt marshes are generally able to withstand violent storms without collapsing and they can therefore be considered resilient to extreme storms [76].

Mangroves likewise demonstrate considerable resilience over timescales of centuries to millennia commensurate with shoreline evolution, including their development during the Holocene [77,78]. Accretion rates in mangrove forests are currently keeping pace with mean sea level rise [79] and mangroves demonstrate resilience in their patterns of recovery from natural disturbances like

extreme storms and tsunamis [80]—traits that put them at the front line of nature-based solutions to mitigating coastal hazards. Indeed, the biggest threat to mangrove systems is not climate change, but deforestation [81].

Tidal wetlands can transition from vegetated platforms to bare tidal flats, or vice versa, as a function of complex feedbacks between water depth, sedimentation, and vegetation patterns [82–85]. These tidal systems tend to eschew intermediate elevations: higher elevations in the intertidal zone tend to support more (and more robust) vegetation that is effective at trapping (and creating) sediment, thus building elevation where elevations are already high. By contrast, lower intertidal elevations experience greater bottom shear stress, which facilitates sediment resuspension and discourages recruitment by colonising vegetation, thus tending to keep low elevations low.

*3.4. Coral Systems*

Biophysical feedbacks in coral island systems also accommodate perturbations from sea level rise and storm events. On long (interglacial) timescales, reef dynamics describe a stable attractor in which coral growth rates adjust as a function of water depth [86]. On shorter, multiannual timescales, island morphology responds to storm impacts through the dynamic reorganisation of motu, the subaerial gravel islands—typically vegetated—atop a reef platform [87], such that island area tends to be conserved or expanded even under conditions of rapid sea level rise [88].

## 4. Resilience and Resistance

Closely associated with resilience—and, by extension, with transitions between alternative stable states—is the concept of resistance. Some consider resistance an intrinsic component of resilience, especially where resistance is a dynamical property derived from traditional engineering and economic ideas about stability [32]. Many geomorphologists, however, consider resilience and resistance to be distinct properties of geomorphic systems [89,90], where resistance is the ability of a geomorphic system to withstand or absorb a change or disturbance with minimal alteration, and resilience is the ability of the system to recover toward its pre-disturbance state [91]. By this definition, resistance is a capacity exerted before the system is perturbed; resilience can be measured after the perturbation has occurred. In geomorphic systems—especially sediment-transport systems—the impacts of physical disturbances can be filtered and disproportionately attenuated (through negative feedbacks), rather than amplified (through positive feedbacks) [92–94]. In some cases, such as in well-developed beach cusps [95] or large-scale cuspate forelands [96] that inhibit the development of smaller-amplitude wavelengths, a negative feedback underpins resilience by reinforcing equilibrium and/or pattern stability [97]—and the presence of the negative feedback itself constitutes a kind of resistance.

When a positive feedback amplifies a perturbation into a change in stable state—for example, when a major disturbance to a vegetated marsh initiates a transition to an unvegetated tidal flat, or when a barrier is breached, converting a freshwater lagoon in an estuarine environment—then the resistance of a system may be overcome, even if it remains ecologically resilient in Holling's [32] typology. Piégay et al. [16] point out a fundamental conflict in this aspect of ecological resilience. Theoretically, a system that crosses a threshold and enters a new state remains resilient and has adaptive capacity because it is composed of living components that can adapt to other environmental conditions. That said, many intrinsic nonliving components may have significantly and/or irreversibly changed. Returning to the example of an intertidal marsh, with a loss of vegetation, high-elevation topography may transition to the low-elevation topography of an intertidal flat. Both conditions are ecologically resilient, but they are fundamentally different environments. They are coupled by a critical dynamical threshold, but nonetheless characterised by their own physical and ecological processes and functions. Returning to the example of a barrier breach, both a freshwater lagoon and an estuary are environments with ecological resilience and high conservation value, but they are vastly different in terms of functioning and biodiversity; consequently, the switch from one environmental state to the other may be unacceptable from some socioeconomic or even conservation point of view.

## 5. Resilience in Coastal Human–Environmental Systems

Social scientists who view communities and societies as socioeconomic systems that can self-organise and function in multiple or alternative equilibrium states describe a view of resilience that is similar to that of ecologists [98,99]. For decades an interdisciplinary branch of resource economics has advanced a theory for coupled social–ecological systems, in which socioeconomic dynamics, among other components, are vital to how a common pool environmental resource system responds to disturbances and shocks [100]. Some scholars consider resilience to have morphological, ecological, and socioeconomic components [101]; others engineering, ecological, community, and social–ecological components [15]; and still others engineering, ecological, and psychological components, where the latter is defined as "the ability of human individuals and communities to withstand and/or recover from disturbances" [22].

Flood and Schechtman [22] argue that recognising, reconciling, and integrating psychology as a primary component of resilience is necessary to capture the complex interplay of human and environmental systems in coastal zones. They propose that increased resilience requires strengthening engineering, ecological, and psychological components in a reinforcing manner, rather than championing one at the expense of others, but such balance is difficult to achieve. For example, the ability of a community to recover psychologically from a devastating coastal storm—to build psychological resilience—may be underpinned by engineering-driven strategies such as infrastructural investment in hard defences, which may in turn weaken ecological resilience [102,103]. Consider the rhetoric of the recovery plan for New York City after Superstorm Sandy in 2012, entitled *A Stronger, More Resilient New York*, which aimed to increase resilience through the building and upgrading of hard engineering defences: "By hardening our coastline . . . we are a coastal city—and we cannot and will not, abandon our water front. Instead we must build a stronger, more resilient city—and this plan puts us on a path to just do that" [104]. This adoption and interpretation of resilience enables the reconstruction of existing communities in the same vulnerable places they existed before the storm, potentially compromising long-term resilience. Similarly, investment in disaster recovery and improved hazard defences might compromise both ecological and psychological resilience—at least for some groups—by catalysing post-disaster gentrification and the displacement of the local pre-disaster community [105,106].

In objective, dynamical terms, a system with more than one stable state may be resilient to perturbations in whichever state it takes. What is not always explicit is a collective preference among those who use and manage a given environmental system for the persistence of one state over any others [1,107]. If coastal resilience is an intrinsic property that arises from the natural ability of coastal systems to adapt to sudden or gradual changes to the drivers of coastal dynamics [101], then the Building with Nature concept [3], for example, represents a deliberate effort to embed these dynamics into management approaches that facilitate resilience in developed and populated coastal zones. This inevitable blurring of natural and built environments—or the outright replacement of natural environments with built ones [1,108]—thus complicates any unified definition of resilience.

Coupled human–environmental systems manifest dynamics that differ substantively from the dynamics of their constituent systems in isolation [103]. The constituent socioeconomic system might describe one attractor, the environmental system another attractor, and the dynamically coupled system still another attractor, distinct from the other two. Consider a city on a delta, like New Orleans. In the absence of any river and coastal flood hazard, the city likely would have evolved to have some other urban structure—hypothetically, a uniform grid—unconstrained by levees. Likewise, in the absence of a city, the Mississippi River, free to distribute sediment across its lowermost floodplains and sustain its coastal marshes, likely would have maintained the elevation of its delta relative to sea level. But combined—a city on a delta—the dynamics of each depend on the other, resulting in hazard-control measures that shape the physical and sociopolitical–economic structure of the city, and changes to the physical geography that amplify hazard [103]. In fact, although some settings are more tightly coupled than others [109], such human–environmental coupling is likely characteristic of all developed coastal

environments. A powerful concept in terrestrial ecology is that the biomes of the world—traditionally defined as natural ecological systems with human systems embedded in them—have changed so fundamentally with human domination of the world's ecosystems [110,111] that they are now anthromes, or human systems with ecological systems embedded in them [108,112]. Invoking global analyses of human impacts on marine and coastal environments by Halpern et al. [113,114], Lazarus [1] has argued that developed coastal environments are so impacted (directly and indirectly) by human activities, from engineering and industry to climate-related change, that the world's coasts now constitute coastal anthromes.

To the extent that modern coupled human–environmental systems are understood, forays into their dynamics tend to be theoretical or compiled from patchworks of case studies [103,115]. In coastal settings, specifically, exploratory numerical modelling suggests that developed coastal barriers with engineered protections against hazard impacts (i.e., chronic erosion, inundation during major storms) exhibit complex dynamical behaviours with distinct attractors, including oscillatory boom–bust cycles in which coastal development intensifies until the costs of protection become unsustainable and the area is abandoned [116–119]. Quantitative empirical tests of this theoretical work, however, are only just emerging [117,120–122].

The variety of possible dynamical attractors for coastal human–environmental systems remains largely unknown. If a boom-and-bust oscillator is potentially one attractor, then a trajectory on that attractor may be the tendency for coastal risk to intensify through a feedback between hazard protection and investment in development [102,103,116,117,120,122–125]. Beyond its promise of short-term financial gain in coastal real estate markets, this is not necessarily a preferred trajectory, or attractor, to be locked into. Other patterns suggest the presence of alternative trajectories, if not alternative attractors. Shoreline management policies such as hold-the-line and managed realignment (typically the abandonment of coastal agricultural land for wetland creation) constitute different dynamical trajectories [2,126], but both are manifestations of a boom-and-bust attractor, as hold-the-line strategies are likely not indefinite and managed realignment may require the deliberate abandonment of pre-existing infrastructure (Figure 2). There are also growing indications that sea level rise is beginning to negatively affect coastal property values in some areas [127]. Economic arguments contend that the preservation of coastal habitats and Building with Nature strategies could ultimately reduce risk and damage costs to coastal infrastructure over timescales relevant to management decision-making [9–12,14].

If management for coastal resilience is interested in the long-term maintenance of a single, stable equilibrium state, then coastal management pursues a general model of engineering resilience. However, imposing a subjective preference for single-state stability onto an inherently multi-state system—that is, forcing the dynamics of ecological resilience to conform to those of engineering resilience—creates a problem of conflicting desires, a case of having cake versus eating it. A preference for stability may be implicit in the management of developed coastal zones, even as the socioeconomic component of the coupled system grows at the expense of its environmental counterpart. Such growth inevitably forces changes in the coupled system in ways that alter its structure, and, by extension, its stability. Given capacity for ecological resilience, the system might adjust to a new stable state—one among perhaps many possible states. By comparison, sustained efforts to maintain a single, preferred equilibrium may ultimately fail. A coupled human–environmental system constrained by engineering resilience and without limits to growth (e.g., [128,129]) is steered toward a state that is increasingly untenable without continuous intervention, such as repeated beach nourishment, and at increasingly large scales [102,103,109,130–132]. In coastal zones likely characterised by a feedback between protection and development, the irony of further investment in coastal protection—an effort to maintain the local steady state—is its indirect stimulus for further development, exacerbating the underlying problem [120,123,133].

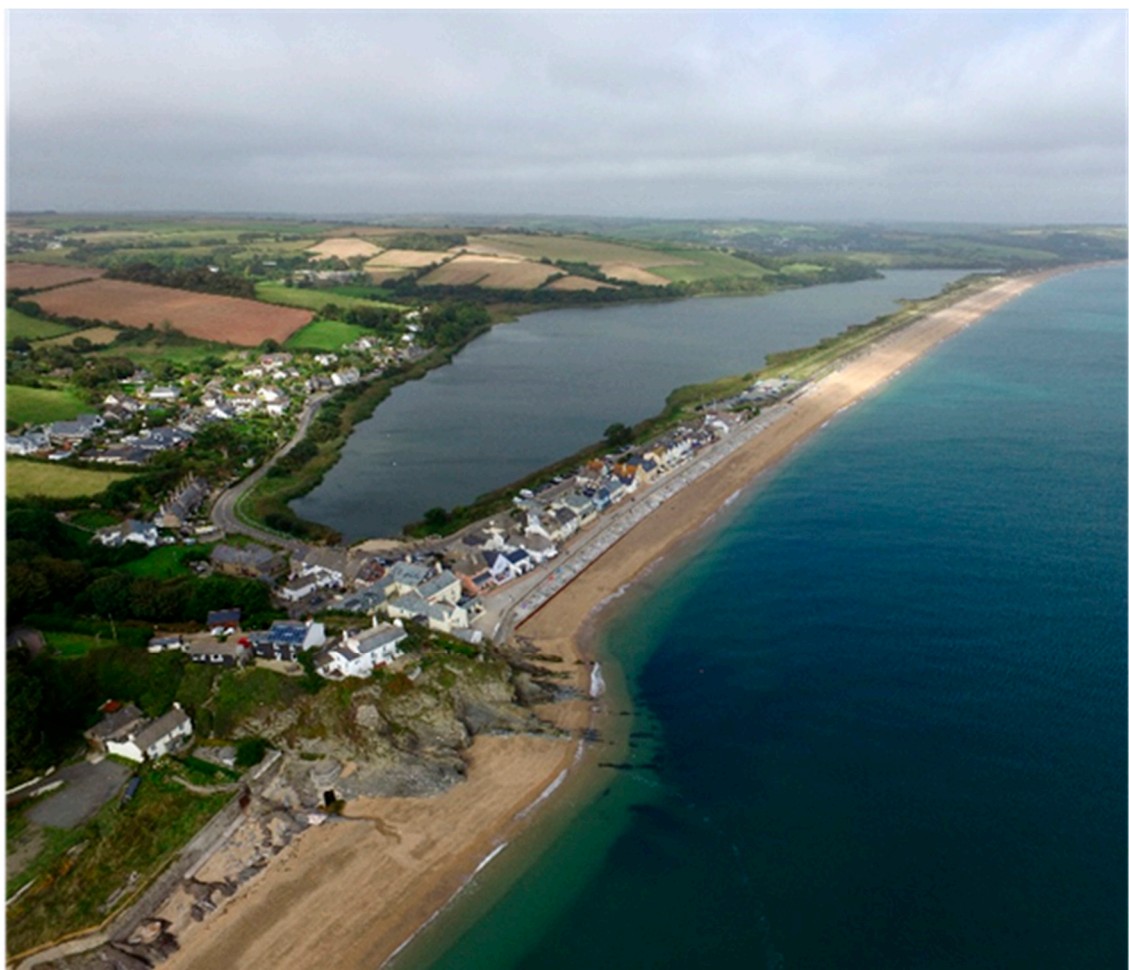

**Figure 2.** The village of Torcross, south Devon, England, is situated at the end of a narrow gravel barrier that separates a freshwater lagoon from the sea (photo: Peter Ganderton). An important road runs along the crest of the barrier. The barrier is highly dynamic and erosion resulting from storms and sea level rise threatens the village and the road. The management policy for the village is hold-the-line, and recent reinforcement of the seawall has undoubtedly contributed to enhanced socioeconomic resilience in the short to medium terms (up to 2050), whilst compromising the natural behaviour of the beach in front to the seawall. The current policy for the road, however, is no active intervention and in case of significant damage to the road it will not be repaired and will thus cease to function. This is likely to have a negative impact on the socioeconomic resilience of the region, but it will allow the barrier–lagoon system to function more naturally, thus enhancing ecological and geomorphological resilience.

Some work has suggested the potential for the incorporation of multiple stable states into restoration programmes for degraded ecosystems [134], and an interesting change is underway in the management of coastal dune systems. Traditionally, coastal dune systems have been restored to, or maintained in, a stabilized state, often through vegetation planting, with the objective to arrest natural geomorphic processes, such as erosion, sediment transport, and dune migration, to improve its role in coastal defence. However, more recent research has shown that dune stabilization can result in the loss of landform dynamics, biodiversity, complexity, and resilience. Artificially stabilized dune systems are often resistant to all but the most extreme disturbances and, as a result, have dysfunctional geomorphic and ecological regimes that do not experience lower magnitude disturbance cycles required for maintaining natural dune ecosystem structure and function [135]. Even well-intentioned interventions can still result in the compartmentalisation of dune landforms and ecologies [136,137]. A management effort that attempts to stabilise a coastline and enhance its resilience may find itself

trying to reconcile contradictory goals [20]. Reestablishment of natural disturbances and related morphodynamics in dune landscapes are being incorporated increasingly into restoration projects that seek to restore lost ecosystem dynamics and services [138–141]. A more dynamic landscape, wherein natural geomorphic processes are stimulated, is argued to provide a more resilient ecosystem with more favourable ecological conditions for native communities and endangered species [142].

Returning reclaimed tidal salt marshes to their natural state is another example of improving degraded ecosystems by restoring their ecological resilience, whilst at the same time enhancing resilience to flooding by increase floodwater storage. Unfortunately, historically impounded marshes can be too low in the tidal frame for salt marsh vegetation to thrive [143]. If starting from an elevation deficit, once-impounded marshes may be less resilient to sea level rise than natural marshes [144]. By contrast, at the mouth of the Yangtze River, abundant sediment load in the system appears to produce resilient reclaimed wetland ecosystems, with wetland development landward and seaward of impoundment structures [145].

## 6. Toward a Working Definition of Coastal Resilience

The generic, widely applied definition of coastal resilience refers to the ability of a coastal system—whether geomorphic, ecological, socioeconomic or a combination [101]—to bounce back from a major shock or disturbance, such as a storm event. Under climate change, however, a more important aspect of coastal resilience is the capacity of a given system to withstand or adapt to a chronic, continuous disturbance, such as sea level rise, a shift in prevailing wave conditions, or a negative sediment budget. An inclusive definition of coastal resilience should therefore account for both types of perturbation—sometimes referred to as pulse versus press/ramp disturbances [16,146].

In addition to recognising different disturbance types, a working definition of coastal resilience should acknowledge the importance of viable function, such as intact sediment transport pathways and physical space to accommodate morphological change and variability. For management purposes, dynamic functionality should perhaps supersede system state: a salt marsh platform might look intact, but in fact be nearing a critical threshold of becoming a tidal mudflat. A restored marsh can have the appropriate vegetation, but if the marsh hydroperiod increases with sea level rise without sufficient sediment input and vertical accretion rates, the marsh is not systemically functional and will likely transition to an unvegetated tidal flat [73,83,84]. The spatial extent over which the intrinsic biophysical feedbacks of tidal wetlands are able to function has a fundamental effect on the variety, integrity, distribution of alternative stable states in the tidal wetland environment at macroscales [41]. A system state is not necessarily a direct indicator of system function. Hence, the essential need for information about both state and behaviour [147].

Over the past two decades, related definitions of coastal resilience have appeared and evolved in the literature of coastal disciplines. The term resilience was first used prominently in relation to coastal zone management and climate change adaptation in the second report of the Intergovernmental Panel on Climate Change (IPCC) [148], and again in the major, international EUROSION project [149]. The latter project framed coastal resilience as: "the inherent ability of a coastline to cope with changes induced by factors such as sea-level rise, extreme events, and human impacts, while maintaining the functions fulfilled by the coastal system over the long-term". The fifth IPCC report defines resilience as: "the capacity of social, economic, and environmental systems to cope with a hazardous event or trend or disturbance, responding or reorganizing in ways that maintain their essential function, identity, and structure, while also maintaining the capacity for adaptation, learning, and transformation" [26].

The 2013 EU strategy on climate adaptation, coastal, and marine issues discusses measures to increase the resilience of European coastlines, maintaining a clear connection between resiliency and integrated coastal zone management [150]. Coastal zone management in the Netherlands, in particular, has embraced a holistic view of resilience [101,151], stating: "The resilience of the coast is its self-organising capacity to preserve actual and potential functions of coastal systems under the influence of changing hydraulic and morphological conditions. This capacity is based on the (potential) dynamics of morphological, ecological and socio-economic processes in relation to the demands that are made by the functions to be preserved."

More sophisticated than traditional definitions derived from simplifications of ecological and engineering resilience, the Dutch definition explicitly recognizes that coastal systems are dynamic and continuously evolving, and that they represent fundamental natural capital for providing and supporting flood protection, recreation, tourism, drinking water supply, housing, and nature conservation. For human welfare, the ecological bases for these functions must be preserved—and that preservation in turn relies on the stewardship of coastal environments. Note that the definition does not prescribe a coastal state that should be aspired to and preserved, but rather the conditions that the coastal system should meet, which provides planners and policymakers with more flexibility [101].

With an eye to these various and overlapping definitions of coastal resilience, we suggest the following synthesis: "Coastal resilience is the capacity of the socioeconomic and natural systems in the coastal environment to cope with disturbances, induced by factors such as sea level rise, extreme events, and human impacts, by adapting whilst maintaining their essential functions."

## 7. From Definitions to Frameworks and Metrics

Beyond definitions for terminology, conceptual frameworks, such as the one developed by [152] for assessing coastal vulnerability, remain relevant for identifying how various systems properties (e.g., susceptibility, resistance, resilience) may be related to disturbance, and for directly addressing the natural and socioeconomic dimensions of modern coastal systems. Resilience and vulnerability tend to be closely associated. Some researchers view the concepts as opposites, arguing that an environment that is vulnerable to a certain stressor (e.g., sea level rise, extreme storms) is not resilient to that stressor [153]; others present them as two sides of the same coin [154]. The framework by Klein and Nicholls [152] exemplifies the latter perspective. In their rendering, susceptibility reflects the potential for a coastal system to be affected by a disturbance (e.g., sea level rise); resistance describes the ability of a susceptible system to avoid or withstand perturbation; and resilience is a measure of the system's capacity to respond to the consequences of perturbation. The natural responses of resistance and resilience are termed autonomous adaptation, in contrast to planned adaptation through human interventions, which can affect coastal resilience by either hampering or enhancing the effectiveness of autonomous adaptation.

For resilience and vulnerability to be applicable concepts that help guide management and inform policy decisions, they ultimately require quantification [155]. Understanding differences in resilience across sites and environments is critical for informing coastal management and policy, but such analysis is hindered by a lack of simple, effective tools. Numerical models can be applied, but these can be complicated and tend to be site specific, making them highly sensitive to parameterisation [156]. The need for relative comparisons—between cases and in a given location over time—has prompted the development of empirically driven indices, such as the Driver-Pressures-State-Impacts-Response (DPSIR) framework [157], the Remote Sensed Resilience Index (RSRI) for coral reef islands [158], and the Coastal Vulnerability Index (CVI) to assess coastal vulnerability to coastal hazards [159–164]. Acknowledging that a single metric for both vulnerability and resilience assessment raises a number of challenges, Lam et al. [165] delivered the Resilience Inference Measurement (RIM), a statistical inferential method that uses real exposure, damage, and recovery data to derive a resilience ranking for a community. As an example of a new approach to characterizing marsh resilience, Raposa et al. [166] developed multi-metric indices for tidal marsh resilience to sea level rise (MARS), incorporating ten

metrics for characteristics that contribute to overall marsh resilience to sea level rise (e.g., percent of marsh below mean high water, accretion rate, tide range, turbidity, rate of sea level rise) and reflect marsh sensitivity and exposure. MARS index scores can inform the choice of the most appropriate coastal management strategy for a marsh—moderate scores call for actions to enhance resilience while low scores suggest investment may be better directed to adaptation strategies such as creating opportunities for marsh migration rather than attempting to save existing marshes.

In coral reef systems, resilience-based management is a rapidly expanding approach in which resilience theory and tools are used to inform decision-making and help set realistic expectations for attainable management goals [167–170]. Assessment of resilience in these coral reef systems is based on the identification and quantification of resilience indicators—a select set of fundamental physical and ecological characteristics that tend to make a reef system more likely to resist and/or recover from disturbances, such as bleaching [171]. Researchers in coral ecosystems are also taking advantage of high-resolution and open-source satellite imagery, and related advances in image analysis, to pioneer new quantitative resilience indicators through remote sensing, such as the Remote Sensed Resilience Index (RSRI) for coral reef islands [158].

Quantifying resilience remains challenging. Salt marshes, for example, have been found to be extremely vulnerable, with large salt marsh losses documented worldwide, and particularly in developed coastal zones [172,173]. At the same time, estimates of critical rates of sea level rise for coastal salt marshes around the world indicate relatively high resilience at many salt marsh sites [174], and all assessments highlight that the available sediment supply is a key factor for marsh resilience to sea level rise [74,75]. Salt marshes in microtidal regimes are particularly sensitive to a reduction in sediment supply under increasing rates of sea level rise, but salt marshes in macrotidal regimes are more resilient to high rates of sea level rise and/or reduced sediment supply [175,176]. Resilience may be an intrinsic property of system structure and interactions, but is nonetheless related to, if not controlled by, site-specific geographical and historical circumstances [91,172,174], further complicating any categorical statements about resilience in geomorphic systems.

Given the critical role that sediment supply plays in the complex dynamics of geomorphic systems, coastal and otherwise, perhaps resilience is, fundamentally, a net-positive sediment budget. As far as single metrics go, the concept is a powerful one. The aim of restoring coastal floodplain connectivity, for example, is to counteract subsidence by allowing floods to rebuild land elevation [14]. Filling out the world's shrinking, sinking deltas will require many kinds of interventions, but none more important than deliberate sediment diversions to build new, compensatory land area [177]. As part of their comprehensive plan to manage their national coastline, the Dutch use a rigorous, systematic programme of beach nourishment to maintain their shoreline at its position in 1990 [178]. A less systematic—and therefore especially surprising—example comes from the eastern seaboard of the United States, where evidence suggests that enough beach nourishment has occurred since the 1960s to effectively reverse the predominant trend of shoreline change from erosion to accretion [122,179].

Even if a single metric for coastal resilience were to exist, it would likely be normalised (imagine a dimensionless index between 0 and 1), and highly sensitive to its constituent components. Consider the closely related concept of risk, defined as a product of hazard, exposure, and vulnerability—hazard is a likelihood that a hazard event of a given magnitude will occur; exposure typically refers to people or infrastructure in harm's way, or to the economic consequences of a hazard impact on infrastructure and livelihoods; and vulnerability is itself a compound metric intended to capture susceptibility to harm from exposure [180–182]. Each component term must reflect the kind of risk being examined and the timescale of consideration. Is the research concerned with punctuated extreme events or chronic flooding and erosion? With numbers of people or numbers of buildings? With demographics or residual economic losses or both, and their interrelationships? The resulting risk index might look the same—a distribution of values between 0 and 1—but its formulation can vary widely. Similarly, a coastal resilience index might hinge on a measure of recovery time to pre-disturbance conditions. But rapid recovery might indicate strong resilience in a beach system—the natural restoration of beach

volume following an erosive storm event [50,183]. But rapid recovery in coastal real estate might have more complicated implications if house prices quickly rebound after a storm event [184]—and serves as another reminder that resilience may convey a preference for one kind of system behaviour over another. Resilience—and therefore any metric for resilience—is context-dependent, but a useful definition of resilience should frame a rich variety of contexts.

## 8. Synthesis and Conclusions

Facilitating coastal resilience is increasingly seen as a desirable outcome for coastal management [185] since a resilient coast is better able to accommodate disturbances driven by natural and anthropogenic processes than one that has limited capacity for internal change [186]. The U.K. Environment Agency strategy for Flooding and Coastal Erosion Risk Management (FCERM) uses "building resilient places" as their objective and vision [2]. Enhancing coastal resilience is increasingly viewed as a cost-effective way to prepare for uncertain future changes while maintaining opportunities for coastal development. Zonation and implementation of buffer zones—reserves, set-back laws, coastal change management areas—should allow the coast to exercise its intrinsic resilience. That said, landform and habitat resilience within coastal human–environmental systems require levels of dynamism and geomorphic complexity not often tolerated by managed systems.

Although resilience is closely linked to dynamical stability, resilient coasts are not necessarily stable coasts. Given that resilience in geomorphic systems is sensitive to local geography and historical legacies [94], blanket conclusions about the relative resilience of particular types of landforms or landscapes (e.g., barrier islands, tidal wetlands, coral atolls) become problematic. And nowhere is the fallacy of stable coasts more important than on developed shorelines. The illusion of stability as resilience enables build–destroy–rebuild cycles of construction and reconstruction of coastal development in hazardous places. Because of the need for rigorous scientific assessments and associated policy implications in vulnerable coastal zones, there is an essential need for clear, consistent definitions and measures of resilience [17].

Coastal environments with an essential ecological component—salt marshes, mangroves, dunes, and coral reefs—perhaps best lend themselves to applications of resilience principles for management. But until the attractors—likely multiple stable states—of coastal human–environmental systems are better understood, managing resilience in anthropogenically dominated contexts will remain a moving target. Moreover, resilience in coastal human–environmental systems will always require a trade-off between the natural environmental and social components, and it is the challenge of coastal management to balance the needs of both the socioeconomic and natural coastal systems for the future, and aim to increase the resilience of both (Figure 3). However, socioeconomic resilience tends to get favoured at the expense of intrinsic natural environmental resilience, such as through the construction of coastal protection structures. Reactive measures that increase resilience across all aspects of the coastal human–environmental system are costly and rare, and perhaps only Building with Nature approaches qualify. There is more scope for proactive measures to enhance resilience within coastal human–environmental systems. A rigorous, science-informed coastal planning approach, implemented at the appropriate temporal scale, remains a feasible tool for achieving proactive adaptation and enhancement of both socioeconomic and natural resilience.

There is no unifying panacea for managing coupled coastal human–environmental systems [187], and pathways to facilitating resilience may not scale easily across local, regional, and national institutions of governance and implementation. What coastal resilience looks like in practice will be diverse, informed not only by physical geography, but also cultural and societal norms.

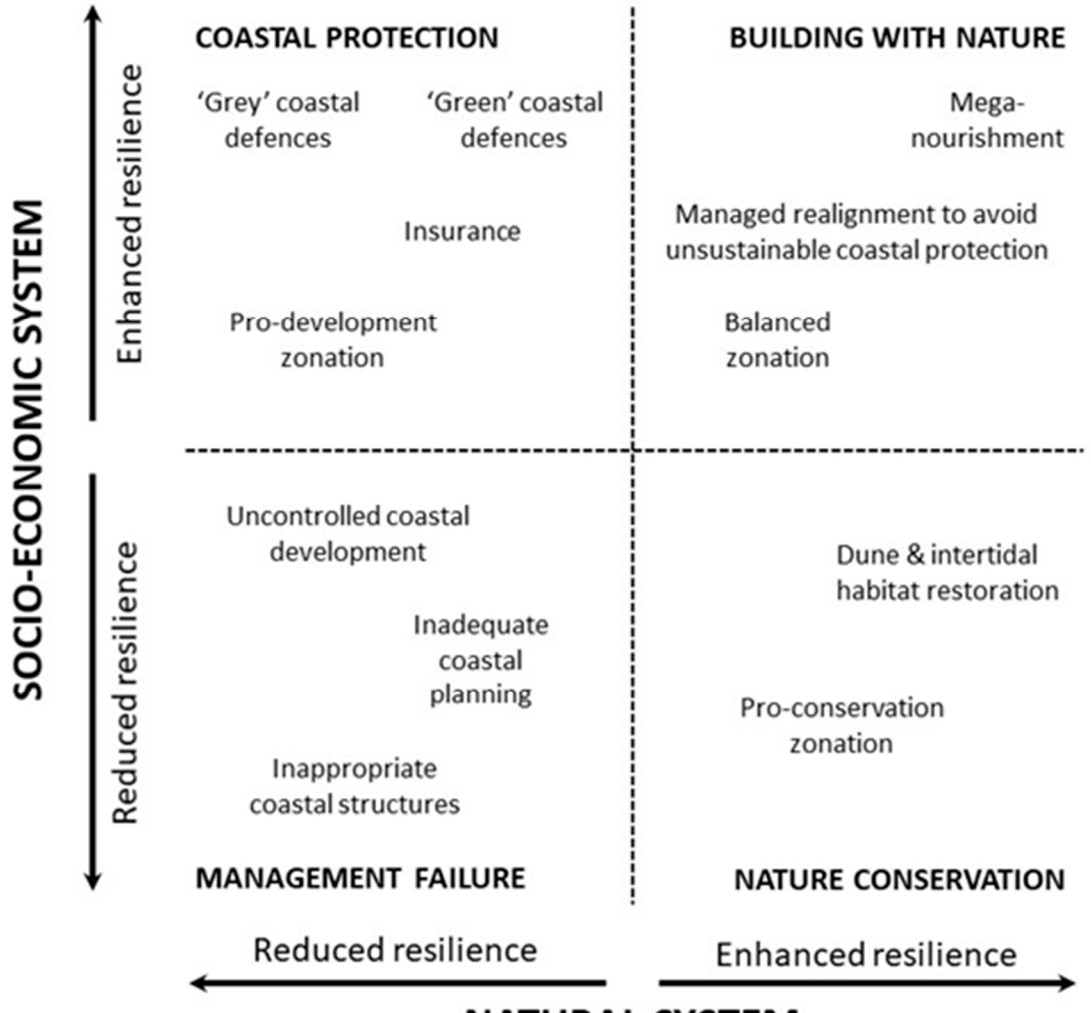

**Figure 3.** Coastal resilience matrix divided into four quadrants and considering the effect of coastal zone management on both socioeconomic and natural resilience. A well-designed and executed mega-nourishment scheme can enhance both socioeconomic and natural resilience (Building with Nature quadrant), while inappropriate coastal structures can have adverse effects on both systems (Management Failure quadrant). Hard engineering structures generally enhance socioeconomic resilience, but almost always reduce natural resilience (Coastal Protection quadrant), whereas pro-conservation measures enhance natural resilience, but this can be at the expense of socioeconomic resilience (Nature Conservation quadrant).

**Author Contributions:** G.M. and E.D.L. contributed equally to all aspects of this paper.

**Funding:** This research was funded in part by the UK Environment Agency (to GM), NERC BLUEcoast project (NE/N015665/1 to GM; NE/N015665/2 to EDL), and the NERC UK Climate Resilience programme (NE/S016651/1 to EDL).

**Conflicts of Interest:** The authors declare no conflict of interest. The funders had no role in the design of the study; in the collection, analyses, or interpretation of data; in the writing of the manuscript, or in the decision to publish the results.

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
