# Peer review of "Defining Coastal Resilience"

_water, doi:10.3390/w11122587_

Round 1

Reviewer 1 Report

A quite atypical revision

It is a very interesting paper that I read with great pleasure. It develops a thought in a coherent and consequential way, also showing the contradictions of a discipline whose theoretical bases have not been properly studied in depth. The fact that forms have a name that implies their genesis, often prevents us from separating the moment of description from that of interpretation. Here we do not fall into this error.

Dealing with feed-back processes in Geomorphology I think that the seminal paper by C.A.M. King should me mentioned (also because all of us is directly or undirectly one of his students!)

King C.A.M., 1970. Feedback relationships in geomorphology. Geografiska Annaler,  n. 52A(3-4), 147-159.

"Builiding with nature" is today's fashion, sensu Sherman (1996) and something could be said in this respect.

Sherman D.J., 1996. Fashion in Geomorphology. In B.L. Rhoades and C.E. Thorn (Eds.), The scientific nature of Geomorphology. Wiley, p. 87-114

In my "very personal opinion", this fashion has also established itself in the not only cultural but also commercial thrust of the Dutch!

You often talk about equilibrium, well explaining that's not a static state.

If the coastal sediment budget is concerned, equilibrium is statistically impossible since too many not feed-back regulated process are working at watershed scale (rainfall, land cover, subsidence, river bed quarryng, levee construction, land reclamation and other human activities,  etc.) and on the sea side (relative sea level, wave climate, biological and chemical processes, etc). If the beach is eroding it cannot send back a message to the river "please, deliver more sediments!".Do you know someone working on this?

For you, and for Water Editor: It's a shame not to be able to read the authors' names when reading an article, and you should stop reading to jump to References. This means that most of the  reader does not know the cited authors. In a paper like this, in which the history of Geomorphology is concerned, names are important.
Try to get around the rules of the journal writing author's name followe by [#], especially at the beginning of a sentence (!). This "fashion" is horrible!

Author Response

NOTE: By our convention, reviewer comments are in italics; our responses are in bold; excerpts of new text are in normal font.

It is a very interesting paper that I read with great pleasure. It develops a thought in a coherent and consequential way, also showing the contradictions of a discipline whose theoretical bases have not been properly studied in depth. The fact that forms have a name that implies their genesis, often prevents us from separating the moment of description from that of interpretation. Here we do not fall into this error.

Dealing with feed-back processes in Geomorphology I think that the seminal paper by C.A.M. King should me mentioned (also because all of us is directly or undirectly one of his students!)

King C.A.M., 1970. Feedback relationships in geomorphology. Geografiska Annaler,  n. 52A(3-4), 147-159.

In Section 4 (regarding "Resilience & resistance"), we now refer to King (1970) as suggested.

Try to get around the rules of the journal writing author's name followed by [#], especially at the beginning of a sentence (!)

We have added author names ahead of the reference numbers where a name begins a sentence, or where the grammatical structure of the sentence uses authors as proper nouns.

Reviewer 2 Report

Defining Coastal Resilience

The title of the manuscript, followed by the material in the abstract, suggest that the author spent considerable paper space conveying the importance of synthesizing a definition of coastal resilience.  The manuscript contains many ideas.  This reviewer commends the co-authors for their efforts along this journey.  The definition appears to be a specialized version of the definition provided by Walker and Salt's (2006, 2012) Resilience Thinking and Resilience Practice books, which is reasonable of course since Walker and Salt deal with social-ecological systems, some along the coast.  The journey for this manuscript is not complete enough for recommending publication as a contribution to knowledge.  The last sentence of the abstract, which includes the term 'optimize', is very telling in this regards, Furthermore, mn manuscript p. 392 “…resilience is a measure of the system's capacity to respond to the consequences of perturbation.”  However, in this statement and those surrounding it in a functional manner, there are no ‘units of measure’ offered as a way to ‘measure’.  The statement requires fleshing out to include the sub-concepts enumerated in the paper, BUT also requires significant synthesis, i.e., identification, integration, and simplification of those sub-concepts into an overall framework of components that contribute to resilience.

Scholarly work preceding an attempt to measure and then optimize resilience would require more analysis and synthesis of concepts involving concerning logical details for specifying the character of resilience than appears in this manuscript. Such work would lay out the functional relationships among all components required for the computational aspects of optimizing resilience emerging out of the definition, but would not necessarily require an empirical study at this time. Such contribution would likely emerge from a collection of synthesized definitions, perhaps embedded within a conceptual framework. The section on conceptual frameworks is moving in the right direction, but is incomplete. The framework would include the most salient sub-concepts, their inter-relationships, plus other related concepts for context.  From there, suggestions for units of measure for all of the sub-concepts are necessary. 

Timothy Beatley (2010) Coastal Resilience, Island Press, is an entire short-book on the topic, but even Beatley never presented a sufficiently robust conceptual characterization that can lead to a computational ‘measurement’ perspective.  

Syntheses in other literatures about resilience could be consulted to provide further insight along the lines of measurement. For example, seismic resilience associated with earthquake disasters is important along coasts. Bruneau et al. (Earthquake Spectra, Volume 19, No. 4, pages 733–752, November 2003, p. 735) offer the following… “…community seismic resilience is defined as the ability of social units (e.g., organizations, communities) to mitigate hazards, contain the effects of disasters when they occur, and carry out recovery activities in ways that minimize social disruption and mitigate the effects of future earthquakes.” They then proceed to deconstruct resilience using four R sub-concepts of…Robustness, Redundancy, Resourcefulness, and Rapidity. Since communities along coasts can be considered social-ecological systems (if any can) and since seismic resilience of coastal communities is important, perhaps there is a good net step there for the authors.

A synthesis of definition sub-concepts is certainly a next step for synthesizing a definition of resilience, even if the manuscript does not carry the process to a ‘logical, functional and measurable’ conclusion. This manuscript does not have it.  As such, with the level of contribution to knowledge stated in the abstract, I conclude that the paper is not conveying enough contribution in 2019, given the state of literature. Continued work is encouraged, perhaps along the lines suggested; but in any case, making the contribution desired requires more work by the co-authors associated with synthesizing concepts and conceptual relations.

Author Response

NOTE: By our convention, reviewer comments are in italics; our responses are in bold; excerpts of new text are in normal font.

The title of the manuscript, followed by the material in the abstract, suggest that the author spent considerable paper space conveying the importance of synthesizing a definition of coastal resilience.  The manuscript contains many ideas.  This reviewer commends the co-authors for their efforts along this journey.  The definition appears to be a specialized version of the definition provided by Walker and Salt's (2006, 2012) Resilience Thinking and Resilience Practice books, which is reasonable of course since Walker and Salt deal with social-ecological systems, some along the coast.  The journey for this manuscript is not complete enough for recommending publication as a contribution to knowledge.  The last sentence of the abstract, which includes the term 'optimize', is very telling in this regards, Furthermore, mn manuscript p. 392 “…resilience is a measure of the system's capacity to respond to the consequences of perturbation.”  However, in this statement and those surrounding it in a functional manner, there are no ‘units of measure’ offered as a way to ‘measure’.  The statement requires fleshing out to include the sub-concepts enumerated in the paper, BUT also requires significant synthesis, i.e., identification, integration, and simplification of those sub-concepts into an overall framework of components that contribute to resilience.

Scholarly work preceding an attempt to measure and then optimize resilience would require more analysis and synthesis of concepts involving concerning logical details for specifying the character of resilience than appears in this manuscript. Such work would lay out the functional relationships among all components required for the computational aspects of optimizing resilience emerging out of the definition, but would not necessarily require an empirical study at this time. Such contribution would likely emerge from a collection of synthesized definitions, perhaps embedded within a conceptual framework. The section on conceptual frameworks is moving in the right direction, but is incomplete. The framework would include the most salient sub-concepts, their inter-relationships, plus other related concepts for context.  From there, suggestions for units of measure for all of the sub-concepts are necessary. 

Timothy Beatley (2010) Coastal Resilience, Island Press, is an entire short-book on the topic, but even Beatley never presented a sufficiently robust conceptual characterization that can lead to a computational ‘measurement’ perspective.  

Syntheses in other literatures about resilience could be consulted to provide further insight along the lines of measurement. For example, seismic resilience associated with earthquake disasters is important along coasts. Bruneau et al. (Earthquake Spectra, Volume 19, No. 4, pages 733–752, November 2003, p. 735) offer the following… “…community seismic resilience is defined as the ability of social units (e.g., organizations, communities) to mitigate hazards, contain the effects of disasters when they occur, and carry out recovery activities in ways that minimize social disruption and mitigate the effects of future earthquakes.” They then proceed to deconstruct resilience using four R sub-concepts of…Robustness, Redundancy, Resourcefulness, and Rapidity. Since communities along coasts can be considered social-ecological systems (if any can) and since seismic resilience of coastal communities is important, perhaps there is a good net step there for the authors.

A synthesis of definition sub-concepts is certainly a next step for synthesizing a definition of resilience, even if the manuscript does not carry the process to a ‘logical, functional and measurable’ conclusion. This manuscript does not have it.  As such, with the level of contribution to knowledge stated in the abstract, I conclude that the paper is not conveying enough contribution in 2019, given the state of literature. Continued work is encouraged, perhaps along the lines suggested; but in any case, making the contribution desired requires more work by the co-authors associated with synthesizing concepts and conceptual relations.

We appreciate R2's remarks, and we agree with many of them, in principle. However, we have been deliberate and careful in the way we have framed this manuscript. The synthesis that we provide here is still a substantive contribution to knowledge, particularly in coastal disciplines, in which the "state of the literature" is anything but consolidated. R2 has essentially described an paper entirely different from the one we present, putting many of R2's suggestions out of scope. (In practical terms, R2 describes not a single paper but a monograph.) Moreover, many of R2's suggestions (e.g., regarding functional relationships between subsystems) are already present in this work, but described in the language of dynamical systems – and we highlight a number of existing frameworks for evaluating resilience (without adding a new one).

To address the spirit, if not the letter, of R2's remarks, we have added the following paragraph at the end of Section 7 ("From definitions to frameworks and metrics"; lines 446–463):

Even if a single metric for coastal resilience were to exist, it would likely be normalised (imagine a dimensionless index between 0 and 1), and highly sensitive to its constituent components. Consider the closely related concept of risk, defined as a product of hazard, exposure, and vulnerability: hazard is a likelihood that a hazard event of a given magnitude will occur; exposure typically refers to people or infrastructure in harm's way, or to the economic consequences of a hazard impact on infrastructure and livelihoods; and vulnerability is itself a compound metric intended to capture "susceptibility" to harm from exposure [180–182]. Each component term must reflect the kind of risk being examined and the time scale of consideration. Is the research concerned with punctuated extreme events or chronic flooding and erosion? With numbers of people or numbers of buildings? With demographics or residual economic losses or both, and their interrelationships? The resulting risk index might look the same – a distribution of values between 0 and 1 – but its formulation can vary widely. Similarly, a coastal resilience index might hinge on a measure of recovery time to pre-disturbance conditions. But, rapid recovery might indicate strong resilience in a beach system – the natural restoration of beach volume following an erosive storm event [50, 183]. But, rapid recovery in coastal real estate might have more complicated implications, if house prices quickly rebound after a storm event [184] – and serves as another reminder that resilience may convey a preference for one kind of system behaviour over another. Resilience – and therefore any metric for resilience – is context-dependent, but a useful definition of resilience should frame a rich variety of contexts.

Also, we now refer to Beatley (2009) in Section 1, line 39.

Reviewer 3 Report

This is a very good contribution to the discussion of coastal resilience. 

The paper is well organised, clear in the narration and concise in most parts. As such I have only minor suggestions:

-While generally focused, on line 74-75 the example given sounds out of place and not necessary, above all because a further example is given later.

-In the same section the example on the cusps is given and it is concluded that dynamical stability is the way that resilience is now formalised. Although I agree with the statement, it is not clear why this is linked to cusps behaviour.

-Section 3.5 analyses the links between the concept of resilience and resistance. This section comes after a series of examples of coastal systems. I found this a bit odd and maybe the authors can find a more prominent location for this section.

-Line 221 there is a reference [2009], I think this is a typo.  

Author Response

NOTE: By our convention, reviewer comments are in italics; our responses are in bold; excerpts of new text are in normal font.

While generally focused, on line 74-75 the example given sounds out of place and not necessary, above all because a further example is given later. In the same section the example on the cusps is given and it is concluded that dynamical stability is the way that resilience is now formalised. Although I agree with the statement, it is not clear why this is linked to cusps behaviour.

We have opted for a kind of compromise here. The preceding sentence (L73–74), "Stability represented the ability of a system to return to an equilibrium state after a temporary disturbance; the more rapid the return, the more stable the system is…", seems to us to need an illustration. Given that R3 did not find the cusps example especially helpful (and R5 also remarks on the example), for clarity we have removed the reference to cusps – particularly because, as R3 notes, plenty of examples are given later.

Section 3.5 analyses the links between the concept of resilience and resistance. This section comes after a series of examples of coastal systems. I found this a bit odd and maybe the authors can find a more prominent location for this section.

We have brought this section into greater prominence by making it a section in its own right (now Section 4). The discussion of resistance makes more rhetorical sense coming after (rather than before) the various coastal environmental examples in Section 3, and serves as a bridge into our examination of human-dominated systems.

Line 221 there is a reference [2009], I think this is a typo.

Fixed, as suggested.

Reviewer 4 Report

I found the paper very interesting dealing on coastal resilience in natural and built coastal environment (mainly bibliographic review).

The title is appropriate and the paper is well-written. Maybe the authors could insert sketches or figures to illustrate the different definitions discussed in the paper.

Author Response

NOTE: By our convention, reviewer comments are in italics; our responses are in bold; excerpts of new text are in normal font.

I found the paper very interesting dealing on coastal resilience in natural and built coastal environment (mainly bibliographic review).

The title is appropriate and the paper is well-written. Maybe the authors could insert sketches or figures to illustrate the different definitions discussed in the paper.

We have added Figures 1, 2 and 3 to illustrate aspects of the concepts we discuss. The captions of the new figures read:

Figure 1 - Gravel barriers are natural forms of coastal defence that protect the hinterland from flooding, whilst at the same time being able to respond to sea-level rise and extreme storms by rolling-back through overtopping and washover processes. They are thus an exemplar of a coastal landform resilient to both pulse and ramp disturbances.

Figure 2 – The village of Torcross, south Devon, England, is situated at the end of a narrow gravel barrier that separates a freshwater lagoon from the sea. An important road runs along the crest of the barrier. The barrier is highly dynamic and erosion resulting from storms and sea-level rise threatens the village and the road. The management policy for the village is hold-the-line, and recent reinforcement of the seawall has undoubtedly contributed to enhanced socio-economic resilience in the short- to medium terms (up to 2050), whilst compromising the natural behaviour of the beach in front to the seawall. The current policy for the road, however, is no active intervention and in case of significant damage to the road it will not be repaired and will thus cease to function. This is likely to have a negative impact on the socio-economic resilience of the region, but it will allow the barrier-lagoon system to function more naturally, thus enhancing ecological and geomorphological resilience.

Figure 3 – Coastal resilience matrix divided into four quadrants and considering the effect of coastal zone management on both socio-economic and natural resilience. A well-designed and executed mega-nourishment scheme can enhance both socio-economic and natural resilience ("Building with Nature" quadrant), while inappropriate coastal structures can have adverse effects on both systems ("Management Failure" quadrant). Hard engineering structures generally enhance socio-economic resilience, but almost always reduce natural resilience ("Coastal Protection" quadrant), whereas pro-conservation measures enhance natural resilience, but can be at the expense of socio-economic resilience ("Nature Conservation" quadrant).

Reviewer 5 Report

Water 631981

Defining coastal resilience

by

Gerd Masselink and Eli D Lazarus

Submitted for publication in Water MDPI

This paper aims to revisit “coastal resilience” as a term and to disentangle its complexities; including conflicting elements such as socio-economic and natural resilience. This paper justifies the need for setting the goal of “optimizing” resilience.

General impression:

The work presented is very interesting as it involves an extremely popular term such as resilience and tries to put it in the correct framework. The reference list is very good as it resists being extremely extensive but it is representative and widely spread among relevant disciplines and academic environments. 

The paper is written in good English.

Should the authors be able to address the comments of this review then I would be able to propose publication.

Major comments

General: The manuscript refers to the possible reasons for temporary disturbance, such as storms. What about less temporary disturbance? I am not referring to sea-level rise (also look at Fröhle et al. 2011) to which the authors refer to clearly. How is coastal resilience related to human activities both inshore (e.g. construction of dams Velegrakis et al. 2008) and nearshore (the construction of new harbours, Eurosion 2004, Uda 2010, Tsoukala et al, 2015)? Indeed, the construction of harbours such as fisheries due to increase food demand or marinas due to coastal tourism is a reality. The former related to the increase of population and the latter to the increase (or the goal to increase) coastal tourism. However, a large discussion is held internationally indicating that although the construction of harbours is meant to enhance tourism and local economy, it very often leads to environmental and socioeconomic failure. Some effort in addressing the above is given in lines 295-304. But clearer indications of examples of direct effect in coastal areas from human activities should be brought forward.

Indeed, with the suggested definition "Coastal resilience is the capacity of the socio-economic and natural systems in the coastal environment to cope with disturbances, induced by factors such as sea-level rise, extreme events and human impacts, by adapting whilst maintaining their essential functions." The authors make clear that “human impacts” is one of the 3 main factors that refer to resilience. Hence more discussion on human impacts should be addressed in the manuscript.

Minor comments:

Lines 76-79: “…with a size determined by the wave conditions and beach slope [32]”. The phrase should be exclude climate change and human activities. Line 346 Please replace “(“ with “{”.

Reference list

Eurosion (2004a) Coastal erosion – Evaluation of the need for action, Living with coastal erosion in Europe: Sediment and Space for Sustainability, PART IV, PART V. (Already referenced in the manuscript)

Fröhle P, Schlamkow C, Dreier N, Sommermeier K (2011) Climate change and coastal protection: adaptation strategies for the German Baltic Sea coast. In: Schernewski G, Hofstede J, Neumann T (eds) Global change and baltic coastal zones. Springer, New York, pp 103–116

Tsoukala V.K., Katsardi V., Hadjibiros K., Moutzouris C.I., 2015, Beach Erosion and Consequential Impacts Due to the Presence of Harbours in Sandy Beaches in Greece and Cyprus, Environmental Processes, 2(1), 55-71

Velegrakis AF, Vousdoukas MI, Andreadis O, Adamakis G, Pasakalidou E, Meligonitis R, Kokolatos G (2008) Influence of dams on downstream beaches: eressos, Lesbos, Eastern Mediterranean. Mar Georesour Geotechnol 26(4):350–371

Uda T (2010) Japan’s beach erosion: reality and future measures. World Scientific, Singapore

Author Response

NOTE: By our convention, reviewer comments are in italics; our responses are in bold; excerpts of new text are in normal font.

General: The manuscript refers to the possible reasons for temporary disturbance, such as storms. What about less temporary disturbance? I am not referring to sea-level rise (also look at Fröhle et al. 2011) to which the authors refer to clearly. How is coastal resilience related to human activities both inshore (e.g. construction of dams Velegrakis et al. 2008) and nearshore (the construction of new harbours, Eurosion 2004, Uda 2010, Tsoukala et al, 2015)? Indeed, the construction of harbours such as fisheries due to increase food demand or marinas due to coastal tourism is a reality. The former related to the increase of population and the latter to the increase (or the goal to increase) coastal tourism. However, a large discussion is held internationally indicating that although the construction of harbours is meant to enhance tourism and local economy, it very often leads to environmental and socioeconomic failure. Some effort in addressing the above is given in lines 295-304. But clearer indications of examples of direct effect in coastal areas from human activities should be brought forward.

Indeed, with the suggested definition "Coastal resilience is the capacity of the socio-economic and natural systems in the coastal environment to cope with disturbances, induced by factors such as sea-level rise, extreme events and human impacts, by adapting whilst maintaining their essential functions." The authors make clear that “human impacts” is one of the 3 main factors that refer to resilience. Hence more discussion on human impacts should be addressed in the manuscript.

We appreciate R5's remarks, and now include (at line 262) a reference (and brief related discussion) to Halpern et al. (2008, 2015) – to date, one of the most comprehensive quantitative assessments of global human impacts in coastal zones.

Halpern, B.S.; Walbridge, S.; Selkoe, K.A.; Kappel, C.V.; Micheli, F.; D’Agrosa, C.; Bruno, J.F.; Casey, K.S.; Ebert, C.; Fox, H.E.; et al. A global map of human impact on marine ecosystems. Science 2008, 319, 948–952. Halpern, B.S.; Frazier, M.; Potapenko, J.; Casey, K.S.; Koenig, K.; Longo, C.; Lowndes, J.S.; Rockwood, R.C.; Selig, E.R.; Selkoe, K.A.; et al. Spatial and temporal changes in cumulative human impacts on the world’s ocean. Nature Commun. 2015, 6, 7615

The new text reads (lines 258–265):

A powerful concept in terrestrial ecology is that the biomes of the world – traditionally defined as natural ecological systems with human systems embedded in them – have changed so fundamentally with human domination of the world's ecosystems [110,111] that they are now anthromes, or human systems with ecological systems embedded in them [108;112]. Invoking global analyses of human impacts on marine and coastal environments by Halpern et al. [113,114], Lazarus [1] has argued that developed coastal environments are so impacted (directly and indirectly) by human activities, from engineering and industry to climate-related change, that the world's coasts now constitute coastal anthromes.

Lines 76-79: “…with a size determined by the wave conditions and beach slope [32]”.The phrase should be exclude climate change and human activities.

In response to R3, we removed the mention of cusps from this paragraph. (That said, we also would remark that the factors raised by R5 do not affect beach cusps at the timescale of their formation. Our sentence is accurate, as written.)

Line 346 Please replace “(“ with “{”.

Fixed, as suggested.

Round 2

Reviewer 5 Report

The authors managed to address the comments made by me and the other 4 reviewers. I am happy to suggest the manuscript for publication.